

# The chemical composition of a new "mica sandwich" foraminiferal species from the East Coast of Korea: *Capsammina crassa* sp. nov.

Somin Lee[1], Eric Armynot du Châtelet[2], Andrew J. Gooday[3], François Guillot[2], Philippe Recourt[2], Fabrizio Frontalini[4] and Wonchoel Lee[1]

[1] Department of Life Science, College of Natural Sciences, Hanyang University, Seoul, South Korea
[2] Univ. Lille, CNRS, Univ. Littoral Cote d'Opale, UMR 8187, LOG, Laboratoire d'Océanologie et de Géosciences, Lille, France
[3] National Oceanography Centre, Southampton, University of Southampton Waterfront Campus, Sothampton, UK
[4] Department of Pure and Applied Sciences, University of Urbino, Urbino, Italy

Corresponding authors
Fabrizio Frontalini,
fabrizio.frontalini@uniurb.it
Wonchoel Lee, wlee@hanyang.ac.kr

## ABSTRACT

We describe a new agglutinated monothalamous foraminiferal species, *Capsammina crassa* sp. nov., based on integrated observations of the test morphology and the chemical characteristics of materials composing the test. The new species was found at a depth of <60 m on the East coast of Korea. The test morphology is typical of the genus *Capsammina*, comprising two or more mica plates with a ring of finely agglutinated mineral grains sandwiched between them and surrounding the cell body. There is no distinct test aperture. Elemental analyses of the agglutinated grains revealed 15 different types of mineral grains of which quartz is the most abundant. The surface areas of grains exposed on fractured surfaces ranged from 1.6 to 7,700 $\mu m^2$ and the large plate-like grains forming the upper and lower surfaces measured about 420–2,350 $\mu m$ in maximum width. The new species is morphologically similar to *C. patelliformis*, however, the differences in size, distribution area and depth support that these two species are distinct. This discovery is the first record of the genus *Capsammina* from the North Pacific. Therefore, it extends the biodiversity and geographical distribution of the genus *Capsammina*, which has been reported only from the bathyal NE Atlantic. Our finding also suggests the possibility of additional discovery of monothalamous foraminifera from around Korea.

# INTRODUCTION

The genus *Capsammina*, and its type species *Capsammina patelliformis* Gooday, Aranda da Silva, Koho & Lecroq, 2010, were established by *Gooday et al. (2010)* based on samples collected at lower bathyal depths in the Nazaré Canyon, off Portugal. This distinctive agglutinated monothalamous foraminifera had been discovered a few years earlier in the

same submarine canyon by *Koho et al. (2007)*, who identified it as *Crithionina* sp. Morphologically, *Capsammina* is characterized by a test that lacks an obvious aperture and by two or more large, plate-like particles of mica forming the upper and lower surfaces and separated by a more or less circular ring of white, fine-grained agglutinated material. Phylogenetically, *C. patelliformis* belongs in a clade branching with several *Crithionina* species but was considered by *Gooday et al. (2010)* to be sufficiently distinct to justify the establishment of a new genus. Additionally, the same authors have suggested that the species *Psammosphaera bowmanni* Heron-Allen and Earland, 1912 may be transferred to *Capsammina* and there is the possibility that *P. bowmanni* may be a variant of *C. patelliformis* based on its use of mica flakes in the test construction and bathymetry. However, more sufficient evidences such as molecular data are required to support these assumptions.

A recent survey based on new and literature data estimated that a total of 818 benthic foraminiferal species, which belongs to 239 genera, 89 families and nine orders, currently inhabit the waters round the Korean peninsula (*Kim et al., 2016*). According to these data, of the orders belonging to the class Monothalamea, only Astrorhizida including seven genera and 11 species have been reported from Korea. The authors suggested that this figure might be an underestimate since samples from deeper waters, as well as monothalamiids, had not been fully considered. Additionally, the middle to north part of the East/Japan Sea, off the Korean peninsula, has not been well studied in comparison with that of the West Coast (Yellow Sea), the South Sea, and the southern part of East/Japan Sea. *Kim et al. (2016)* concluded that further sampling in these understudied areas, as well as at greater depths, would certainly lead to an increase in the total known foraminiferal species diversity around the Korea peninsula. Support for this prediction came in May 2016 with the recognition of a new *Capsammina* species at a site off the eastern coast of Korea, only the second record of this genus outside the Iberian, Celtic and Scandinavian margins of western Europe. The purpose of the present paper is to describe this new species based on test morphology, including a detailed analysis of the chemical and mineral composition of the test.

## Study area

The East/Japan Sea surrounds the Korean peninsula, Japanese archipelago and the Russian territory of the Asian continent (Fig. 1). This Sea has the characteristics of a semi-enclosed, marginal basin and is connected to the adjacent East China Sea, northwestern Pacific Ocean and Okhotsk Sea through four shallow straits, Korea (Tsushima), Tsugaru, La Perouse (Soya), and Tartar, which together have a maximum depth is about 130 m (*Ozawa, 2003*; *Oba & Irino, 2012*; *Khim & Bahk, 2014*). The waters of the East/Japan Sea increase sharply in depth from the coast (*Yu et al., 2011*), averaging approximately 1,500 m with a maximum depth of about 3,700 m (*Choi et al., 2012*). The eastern coast of the Korean peninsula is influenced by the North Korea Cold Current (NKCC) and the East Korea Warm Current (EKWC) (Fig. 1). The NKCC, characterized by cold and low salinity water, flows from north to south along the east coast of the Korean Peninsula. The EKWC breaks out from the Tsushima Warm Current and characterized by a higher

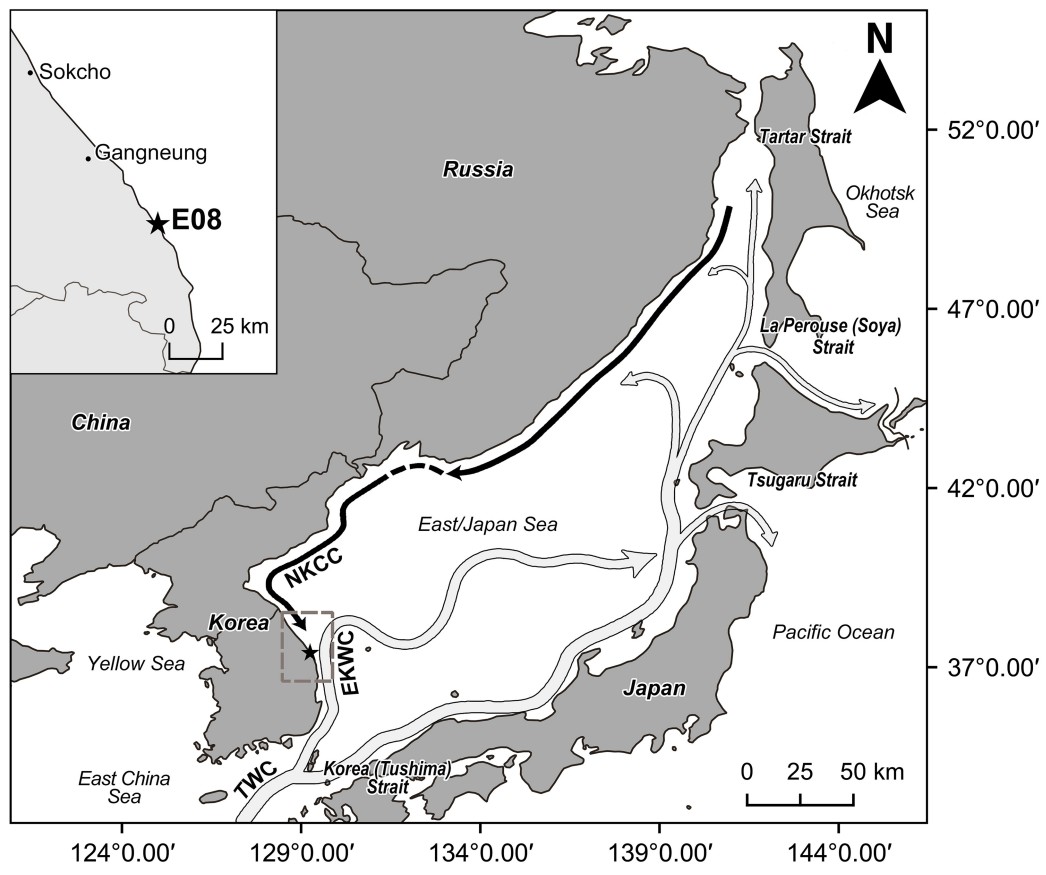

**Figure 1** **The map of study area and location of sampling site (station E08, 37º23′55.665″N, 129º14′ 57.671″E) marked with a star.** Arrows indicate schematically the water currents of the East/Japan Sea (dark arrow: cold water current, light arrow: warm current). EKWC separates from TWC and flows northward. NKCC flows southward along the east coast of Korean Peninsula. EKWC = East Korea Warm Current. NKCC = North Korea Cold Current. TWC = Tsushima Warm Current. This map is made with Natural Earth. Free vector and raster map data at http://naturalearthdata.com/, and QGIS software v.2.18.18, a free and open source geographic information system. Water current information are from schematic map of surface current in the neighboring seas of Korea by Korea hydrographic and oceanographic agency.             

temperature and salinity. It also flows from the south to the north along the east coast of the Peninsula and merges with the NKCC at around 37°–38°N latitude to form a subpolar Frontal Zone characterized by high productivity and rich biodiversity (*Ashjian et al., 2005*; *Lee, Park & Kim, 2016*; *Yoon et al., 2016*). The East Sea water is represented by four water masses—Tsushima Surface Water, Tsushima Middle Water, North Korean Cold Water, East Sea Proper Water—defined by water temperature, salinity, dissolved oxygen and water depth (*Yoon, Jung & Yoon, 2007*; *Choi et al., 2012*). These water masses are strongly influenced by seasonal variations of environmental factors; as a result, the intensity and range of current flow change dynamically and seasonally (*Cho & Kim, 2000*; *Furey & Bower, 2005*). A unique water mass structure is formed in summer due to the south-east monsoon wind, which intensifies the flow toward the north. In winter, the northwest monsoon wind causes the water flow to become stronger southwards (*Yoon, Jung & Yoon, 2007*).

The new species was discovered on May 21, 2016 at station E08 of the "Korean National Marine Ecosystem survey" (Fig. 1). The water depth at this site was 54 m. The bottom-water temperature was 5.03 °C, lower than the mean value (5.85 ± 3.34 °C), while salinity was 33.99 PSU, close to the mean value (33.83 ± 1.15 PSU) from the 23 east coast sampling stations of the "Korean National Marine Ecosystem survey" (*Korea Marine Environment Management Corporation (KOEM), 2016*). Since Station E08 is located within the mixing zone between NKCC and EKWC (Fig. 1), the temperature and salinity values were not significantly different from the mean values. At station E08 the grain size was dominated by silt-clay (5.57ϕ), close to the mean value (5.47 ± 2.90 ϕ), and the organic carbon content was 0.78%, lower than the mean value of 1.36% ± 0.75% derived from all 23 stations (*Korea Marine Environment Management Corporation (KOEM), 2016*).

## MATERIALS AND METHODS

Sediments at station E08 were collected with a Van Veen grab. Immediately after sampling, the uppermost layer of sediment (ca. one cm) was stored in 250 ml bottles and treated with an ethanol-rose Bengal solution (2 g/l) to distinguish between living and dead foraminiferal specimens. In order to obtain additional specimens to analyze, the same station was sampled again on May 16, 2018 and sediment samples immediately frozen at −80 °C.

In the laboratory, each sediment sample was divided into two aliquots, one for qualitative study and the other for quantitative study of benthic foraminifera. Each aliquot was placed in 50 ml bottles and dried in an oven at 40 °C. After the sediments were completely dried, each aliquot was weighed. Aliquots for qualitative study were then gently washed through a 63-μm-mesh screen with tap water to remove clay, silt and any excess dye. The sieve residues were re-dried at 40 °C and weighed to determine, by difference, the mud fraction. Benthic foraminifera were picked using a wet brush from the dried residues under an Olympus SZ40 dissecting microscope. Digital photographs of the new species were taken using an Olympus PEN Lite E-PL3 camera attached to an Olympus SZX12 dissecting microscope. Selected specimens were mounted on a stub, coated with Au-Pd and examined in a FEI Quanta 200 scanning electron microscopy (SEM).

The maximum and minimum dimensions of 20 selected tests, including those of the mica plates and the agglutinated rings, were measured using the Axiovision Rel. 4.8. program. This program was also used to measure the maximum and minimum thickness of the agglutinated ring and the test in lateral view. In addition, the area of each mica plate was calculated using the ImageJ program.

Six individuals of the new species were selected for detailed analysis of the size distribution of the mineral grains and their chemical compositions. The size of the grains was evaluated by means of image analysis using the software Fidji and R on the flat surface that was exposed by the removal of one of the large plate-like mineral grains that define the shape of the test. After threshold adjustment and binarization of the image, grains were separated prior to being counted using watershed

transformation tools (https://imagej.net/Interactive_Watershed). The area of each grain was automatically determined and these data used as the basis for statistical treatment.

The methods used for the determination of the chemical composition of the mineral grains agglutinated in the foraminiferal tests largely followed the ones of *Armynot du Châtelet et al. (2013b)*. All the observations were performed by an environmental scanning electron microscope with an energy-dispersive spectroscopy device FEI Quanta 200 (ESEM-EDS). This technique is directly applicable on unpolished foraminiferal surface and allows us to determine the grain shapes and to identify their mineralogical composition. The specimens were positioned on stubs and carbon-coated to improve the quantification of chemical elements and to produce quality pictures. The EDS data allow a quantification and qualification of the chemical elements distributed within the surface of the agglutinated test (*Armynot du Châtelet et al., 2013a*, *2013b*; *Armynot du Châtelet, Frontalini & Guillot, 2014*). As a first step, the foraminiferal tests were visualized using secondary electrons to determine their general shape, and then, as a second step, imaged for mapping selected chemical elements, namely S, Al, S, Cl, Fe, I, Ti, Ca, Mg and K. The measurement and mapping were carried out using a 20 kV beam. An image analysis based on the chemical maps allows us to characterize the mineral diversity whereas the mineral nature was constrained on point analyses' results.

In order to visualize the distribution and elemental composition of the mineral grains, specimens were reconstructed as 3D models from SEM images using SEM photogrammetric techniques (*Eulitz & Reiss, 2015*). Specimens were glued at the end of a 50-µm-thick copper wire and carbon coated. Within the chamber of the SEM, specimens were rotated to get surface (backscattered electron) and chemical (secondary electron) images every 20°. Photogrammetric reconstruction of 3D models was carried out using Visual SFM software.

The electronic version of this article in portable document format will represent a published work according to the International Commission on Zoological Nomenclature (ICZN), and hence the new names contained in the electronic version are effectively published under that Code from the electronic edition alone. This published work and the nomenclatural acts it contains have been registered in ZooBank, the online registration system for the ICZN. The ZooBank life science identifiers (LSIDs) can be resolved and the associated information viewed through any standard web browser by appending the LSID to the prefix http://zoobank.org/. The LSID for this publication is: urn:lsid:zoobank.org:pub:3DDBFCFC-5927-41C4-9BB6-DDDD39DC22ED. The online version of this work is archived and available from the following digital repositories: PeerJ, PubMed Central and CLOCKSS.

## SYSTEMATIC DESCRIPTION

Supergroup Rhizaria Cavalier-Smith 2002

Phylum Foraminifera Cavalier-Smith 1998

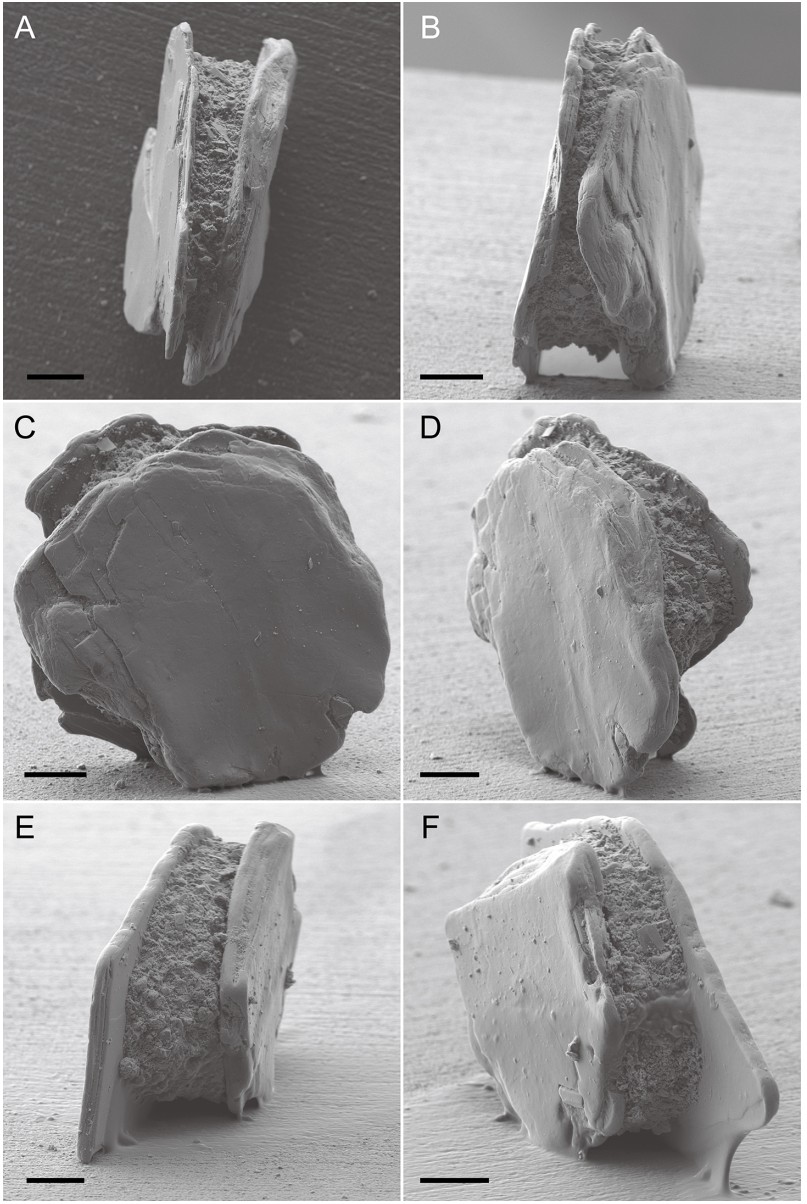

**Figure 2** *Capsammina crassa* **sp. nov. secondary electron images of two specimens (A–F).** The two large micas form the exterior of the test, largely obscuring the small grains that form the inner wall. Scale bars = 100 µm.                               

Class Monothalamea Haeckel, 1862

Order Astrorhizida Lankester, 1885

Genus *Capsammina* Gooday, Aranda da Silva, Koho & Lecroq, 2010

*Capsammina crassa* sp. nov. (Figs. 2–8; Tables 1–2)
urn:lsid:zoobank.org:act:712CADFE-DC11-4BA6-BDD4-A0EB1D80343D.

**Etymology:** The name "crassa" is derived from the Latin "crassus" meaning "fat, big, thick" referring large mica plate of new species.

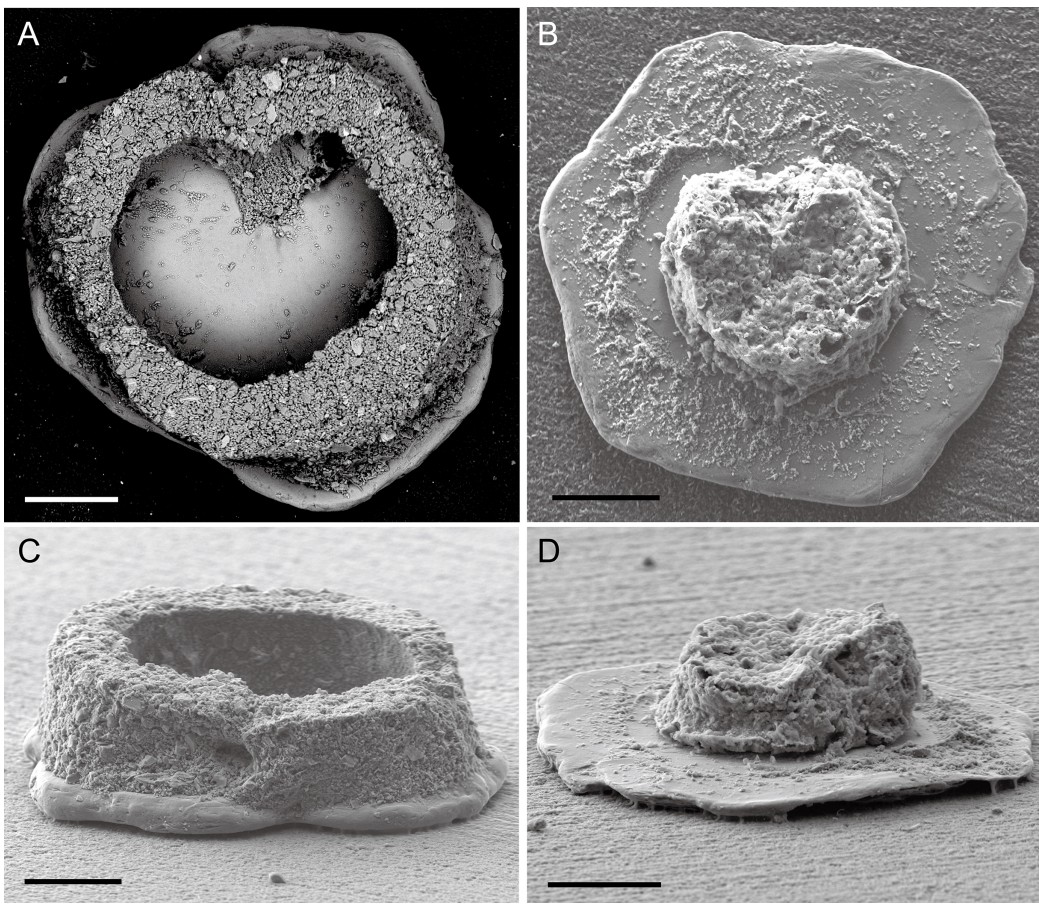

**Figure 3** *Capsammina crassa* **sp. nov.** Test of after removing one of the mica plates. The frontal and side view of the test with one mica plate removed are shown in A and C. The organic remains of the cell are visible in images B and D. Scale bars = 100 μm.

**Diagnosis:** Relatively large species of *Capsammina* characterized by compressed, sandwich-like construction with two flat mica plates attached to upper and lower sides of ring composed of finely agglutinated particles ring test surrounding cell body. Maximum size of test, including mica plates, up to ~2,350 μm (occasionally more); maximum diameter of agglutinated ring up to ~1,260 μm.

**Type material:** Holotype: Fig. S1. C, registration no. NIBRPR0000109500. Paratypes: five specimens, Figs. S2. A, B, D–F, registration nos. NIBRPR0000109501—5, additional 14 specimens, registration nos. NIBRPR0000109506—19. All specimens on micropaleontology slides from "Korean National Marine Ecosystem Survey" Station E08; 37°23′55.665″N, 129°14′57.671″E, water depth 54 m, were collected on May 2016 and 2018. Holotype and paratypes are deposited in the National Institute of Biological Resources, Korea. The additional paratypes are deposited in the Marine Biodiversity Institute of Korea.

**Other examined material:** The mineral grain-size distribution and chemical composition of six specimens from the type locality were analyzed.

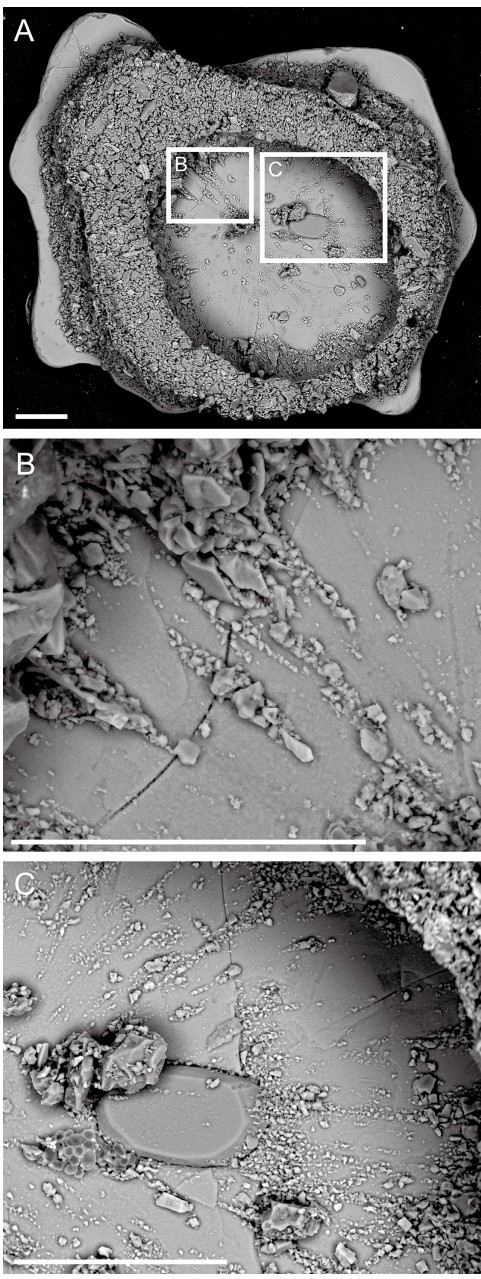

**Figure 4 *Capsammina crassa* sp. nov.** Backscattered SEM image showing smaller grains arranged in radiating linear patterns, probably corresponding to pseudopodia and possibly associated with test construction. (A) Full frontal view of a test without one of the mica plates. (B and C) Close-up of smaller grains arranged inside the test shown in (A). Scale bars = 100 μm.

## DESCRIPTION

*Test structure*. The test is unilocular and lacks an obvious aperture. It is compressed with two irregularly-shaped, plate-like, mineral particles forming the flat top and bottom sides of test (Fig. 2). One of the plates is transparent and the other opaque, the transparent plate typically being the same or smaller in size than the opaque plate. The two plates are separated by a whitish, circular to broadly oval ring composed of agglutinated mineral

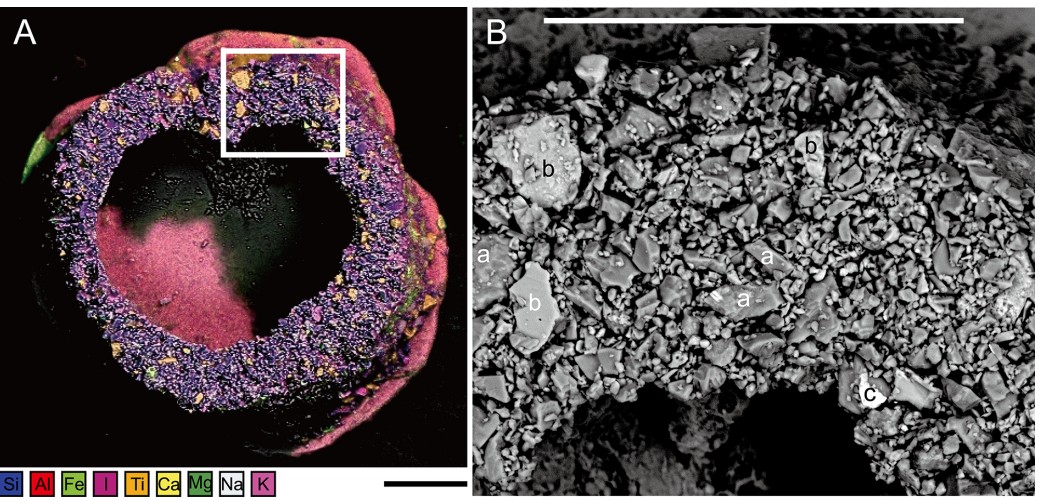

**Figure 5 *Capsammina crassa* sp. nov.** (A) ESEM-EDS image showing elemental composition of the agglutinated grains; the following elements are distinguished: Si, Al, Fe, I, Ti, Ca, Mg, Na and K. (B) Secondary electron image showing the density contrast between the quartz grains (a) mainly composed of Si (darker), the calcite grains (b) mainly composed of Ca (intermediate gray) and the oxides (c), in this case containing Fe composition (bright grains). Scale bars = 100 μm.

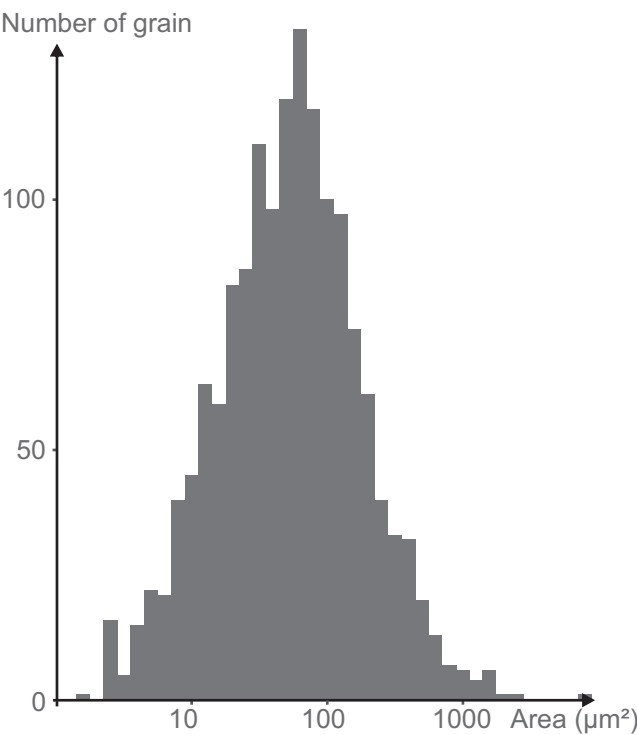

**Figure 6 Log-normal distribution of the grain size of the minerals attached to the mica plate.**

particles that forms a wall enclosing the cell body. The constituent grains are quite variable in size but generally small, appearing finely granular at low magnifications. The coherence between the plates and the finely agglutinated test wall is relatively weak, so that

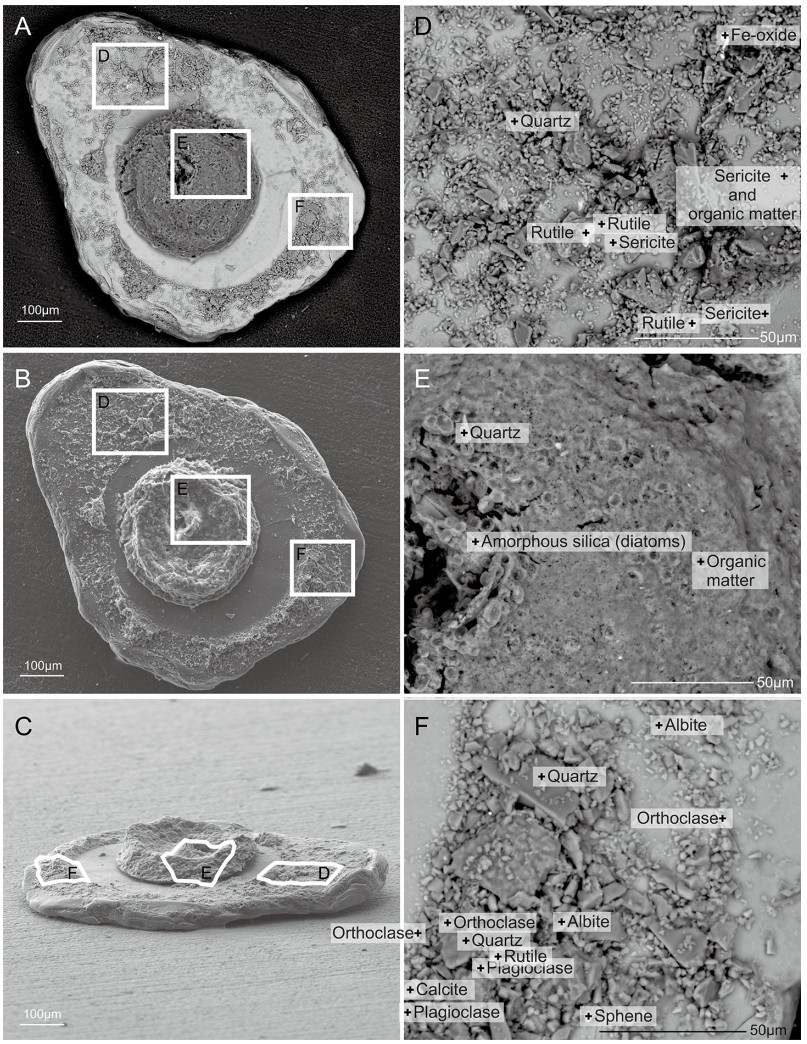

**Figure 7 *Capsammina crassa* sp. nov.** Secondary electron SEM of the test interior showing the underside of the large mica plate, the base of the finely-agglutinated inner wall, and the organic remains of the cell in the center (A–C). Three detailed images show the mineralogical diversity of the grains in the wall (D and F) and details of the cell body containing diatoms (E).

these two components of the test can be separated fairly easily (Figs. 3 and 4). Examination of the underside of some plates exposed in this way reveals the presence of some smaller grains stuck onto the surfaces by very thin layers of organic material organized as an elongated network radiating from the center of the specimen towards the border (Fig. 4).

The maximum dimension of the test (in effect, the mica plates) is generally between 421 and 2,348 μm (average of 956 ± 425 μm) and the minimum dimension between 243 and 992 μm (average of 549 ± 162). The agglutinated ring ranges from 443 to 1,264 μm maximum dimension (average of 851 ± 197 μm) and the minimum dimension from 391 to 1,032 μm (average of 656 ± 157 μm). It is 33.7–456 μm thick, with the thickness generally varying somewhat around the circumference of the ring. In detail, the maximum thickness of the agglutinated ring varies from 108 to 456 μm (average of 235 ± 85.8 μm),

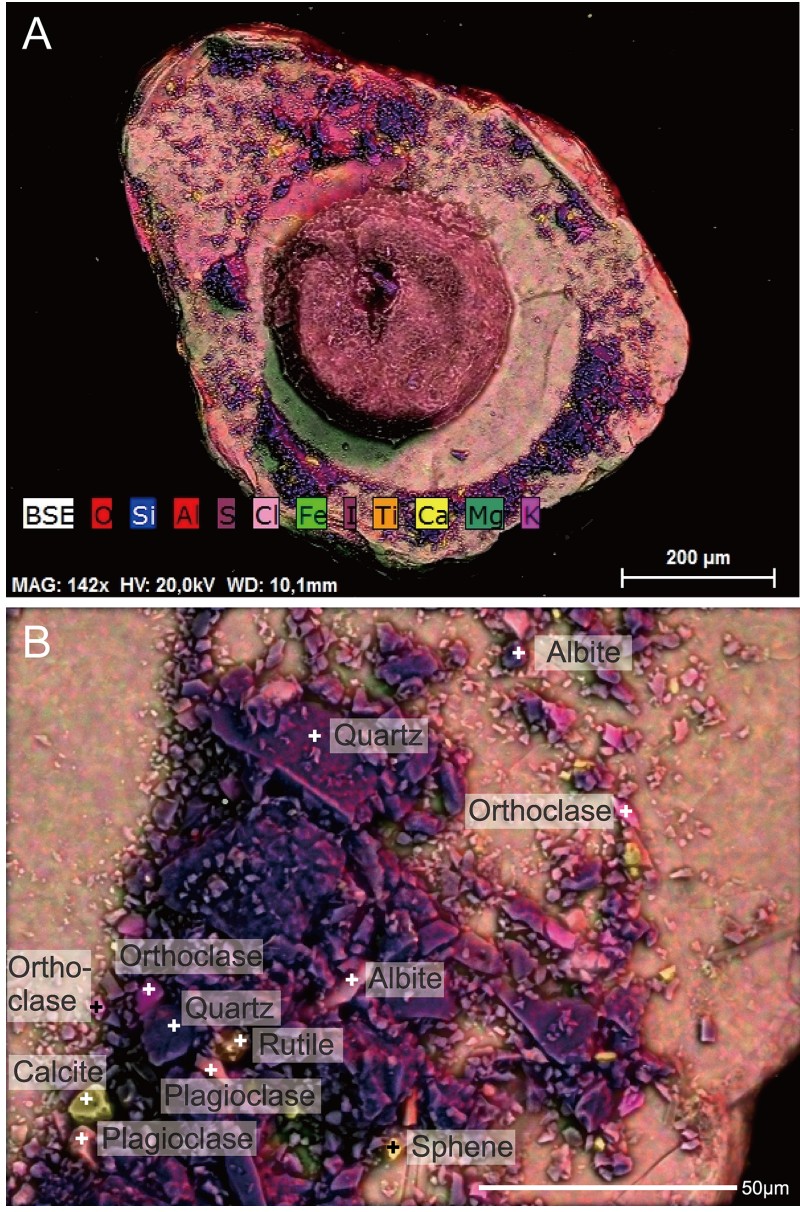

**Figure 8** *Capsammina crassa* **sp. nov. ESEM-EDS color image of the interior of the specimen illustrated in Fig. 7.** Individual mineral grains are identified based on point elemental analyses. (A) The underside of the lmica plate in Fig. 7, showing the base of the finely-agglutinated inner wall and the organic remains of the cell in the center. (B) Close-up of (A) showing the types of minerals located at '+' shaped points.

and the minimum thickness varies from 33.7 to 328 μm (average 111 ± 70.3 μm) (Table 1). The mica plate area of measured individuals varies from 83,759 to 1,250,947 μm$^2$ (average of 410,162 ± 210,548 μm$^2$) (Table 1). One paratype (Fig. S1A) is somewhat larger, the test including the mica plates measuring 1,516 by 992 μm, and the more or less circular agglutinated ring, 994 by 891 μm in diameter and 67–175 μm thick. In side view, the mica plates are not always exactly parallel and so the width between two plates is not uniform. The maximum thickness in lateral view varies from 70.3 to 349 μm
**Table 1 Summary metrics (in μm) for the 20 measured individuals.**

|  | Minimum (μm) | Maximum (μm) | Average (μm) | Standard deviation (μm) |
|---|---|---|---|---|
| max. dimension | 421 | 2,348 | 956 | 425 |
| min. dimension | 243 | 992 | 549 | 162 |
| max. ring dimension | 443 | 1,264 | 851 | 197 |
| min. ring dimension | 391 | 1,032 | 656 | 157 |
| max. ring thickness | 108 | 456 | 235 | 85.8 |
| min. ring thickness | 33.7 | 328 | 111 | 70.3 |
| plate area ($\mu m^2$) | 83,759 | 1,250,947 | 410,162 | 210,548 |
| max. lateral thickness | 70.3 | 349 | 177 | 75.6 |
| min. lateral thickness | 30.7 | 183 | 96.6 | 48.5 |

Note:
max, maximum; min, minimum.

**Table 2 Summary metrics (in μm) for five individuals composed of three mica plates and 15 with two mica plates.**

|  |  | Minimum (μm) | Maximum (μm) | Average (μm) | Standard deviation (μm) |
|---|---|---|---|---|---|
| max. dimension | three-plated (five individuals) | 562 | 2,285 | 938 | 447 |
|  | two-plated (15 individuals) | 421 | 2,348 | 930 | 350 |
| min. dimension | three-plated (five individuals) | 259 | 736 | 478 | 154 |
|  | two-plated (15 individuals) | 243 | 992 | 585 | 157 |
| max. ring dimension | three-plated (five individuals) | 717 | 1,083 | 938 | 147 |
|  | two-plated (15 individuals) | 443 | 1,264 | 822 | 207 |
| min. ring dimension | three-plated (five individuals) | 621 | 919 | 705 | 122 |
|  | two-plated (15 individuals) | 391 | 1,032 | 640 | 168 |
| max. ring thickness | three-plated (five individuals) | 189 | 386 | 274 | 83.9 |
|  | two-plated (15 individuals) | 108 | 456 | 222 | 85.2 |
| min. ring thickness | three-plated (five individuals) | 74.9 | 266 | 140 | 76.2 |
|  | two-plated (15 individuals) | 33.7 | 328 | 101 | 68.1 |
| plate area ($\mu m^2$) | three-plated (five individuals) | 116,997 | 690,741 | 353,254 | 191,942 |
|  | two-plated (15 individuals) | 83,759 | 1,250,947 | 438,616 | 216,707 |
| max. lateral thickness | three-plated (five individuals) | 104 | 290 | 191 | 67.6 |
|  | two-plated (15 individuals) | 70.3 | 349 | 172 | 79.8 |
| min. lateral thickness | three-plated (five individuals) | 30.7 | 105 | 61 | 27.7 |
|  | two-plated (15 individuals) | 35.9 | 183 | 109 | 48.6 |

Note:
max, maximum; min, minimum.

(average 177 ± 75.6 μm), and the minimum thickness varies from 30.7 to 183 μm (average 96.6 ± 48.5 μm) (Table 1). Five of the 20 measured individuals include three mica plates. Detailed test measurements for specimens with two and three mica plates are given in Tables S1 and S2 and summarized in Table 2. In general, specimens with two mica plates reach greater maximum dimensions than those with three plates.

***Granulometry, chemistry and mineralogy of test particles***. The grains constituting the wall (excluding the mica plates) have exposed areas ranging from 1.6 to 2,000 $\mu m^2$ with a

few grains up to 7,700 $\mu m^2$ (Fig. 5), yielding a close to log-normal distribution (Shapiro-Wilk test on log of the grain size: $p$-value = 0.08) (Fig. 6). The fine grains represent 61% of the exposed surface when one of the mica plates is removed, and the remaining area is occupied by organic matter that cements the grains and secures them to the plates.

A total of 70 points analyses of the elemental composition of individual grains was carried out on the six specimens examined using ESEM-EDS (Table S3). The analyses included as wide a range of grains as possible, including the larger plate-like particles covering the specimens and much smaller ones constituting the main test wall. After equilibration of the chemical formulas of the mineral grains, 38 analyses correspond to clearly identified minerals, 26 correspond to minerals but with some doubt regarding their identification, five are of organic composition, and one grain yielded no chemical data. The following minerals were recognized (Figs. 5, 7 and 8): albite, biotite, calcite, dolomitic calcite, $Fe(OH)_3$, orthoclase, oxychlorite, phengite, plagioclase, quartz, rutile, sericite, sphene (titanite) and glass. Quartz grains are dominant in terms of number (Figs. 5A and 8B; blue–violet color). The large plate-like grains forming the upper and lower surfaces of the test are either sericite or phengite (members of the mica family). Some of the layers of sericite are altered to oxychlorite (Figs. 7 and 8).

The 3D reconstruction of the test and its chemical composition is shown in Video S1 and S2. A complete 3D chemical reconstruction was not possible because of the large mica grains create shadows that limit the photons reaching the EDS detector (Video S3). Despite this, the different chemical composition of the two micas and the large-scale distribution of quartz grains are both clearly visible.

*Cell body*. The cell body is clearly visible through the transparent mica plate and occupies most (~85%) of the diameter of the lumen (Figs 3 and 7; Fig. S1). SEM images reveal the presence of diatoms, presumably food items, within the cytoplasm (Fig. 7E).

**Remarks:** *Capsammina patelliformis*, the type species of genus *Capsammina*, was the only species belonging to the genus (*Gooday et al., 2010*). *C. patelliformis* was described from the Portuguese margin in the NE Atlantic Ocean. It closely resembles the new species described here in having a distinctive, sandwich-like test that incorporates two large mica plates. The main morphological difference is the larger size of the new Korean species. The test, including the mica plates, is up to 2,348 $\mu m$ in maximum dimension, while the inner ring of finely agglutinated particles is up to 1,264 $\mu m$ in diameter. The corresponding dimensions for *C. patelliformis* are 640 and 260 $\mu m$, respectively. The width of the test as seen in side view is greater in *C. crassa*, up to 349 $\mu m$ compared to a maximum of 80 $\mu m$ in *C. patelliformis,* and the mica plates are thicker (compare *Gooday et al., 2010*, Pl. 4A with Fig. 2 of the present paper). Finally, although *Gooday et al. (2010)* did not give precise measurements for the agglutinated grains constituting the inner part of the test of *C. patelliformis*, it appears from their published SEM images (e.g., Pl. 4, Fig. A in *Gooday et al., 2010*) that these particles are somewhat larger than those we observed in the new species (Fig. 5B).

Unfortunately, it proved impossible to amplify DNA from any of our Korean specimens and so we are unable to confirm that *C. crassa* is genetically distinct from *C. patelliformis.*

However, the fact that they originate from localities more than 10,000 km apart, and at water depths differing by up to more than 3,000 m, persuades us that the relatively minor differences noted above are sufficient to regard the Korean and NE Atlantic species as distinct. *Gooday et al. (2010)* assigned additional specimens found at shallower sites in the Nazaré (2,847, 1,160, 927 and 344 m depth) and Whittard (2,436, 1,389 m) canyons to *C. patelliformis*. However, at least some of these tended to have more three-dimensional tests (*Gooday et al., 2010*, Pl. 5) compared to those from the type locality. Since no genetic data were obtained from them, it is possible that these shallower specimens represent a different species. The comparison should therefore be between *C. crassa* and *C. patelliformis* from the type locality (3,565 m depth) and nearby sites at similar depths (~3,500 m) in the Nazaré canyon (*Gooday et al., 2010*, Table 1). *C. crassa* sp. nov. occurs depths less than 100 m on the east coast of Korea, in the North Pacific Ocean. Since the East/Japan Sea is a semi-enclosed marginal sea, direct interactions between the two habitats are extremely unlikely.

The species *P. bowmanni* resembles *C. crassa* in having mica plates exposed on the test surface and no distinct aperture. *P. bowmanni* was originally described by *Heron-Allen & Earland (1912)* from the Firth of Forth (Scotland) and subsequently found elsewhere in Scotland (*Murray, Alve & Cundy, 2003*) as well as Scandinavia (*Höglund, 1947*; *Alve, 1990*). There are also records from New Zealand waters (*Höglund, 1947*; *Dawson, 1992*) and Laptev Sea, part of the Arctic Ocean (*Lukina, 2001*). Both species also have similar bathymetric distributions; *P. bowmanni* occurs at 55–66 m water depth at its Scottish type locality, and *C. crassa* at 54 m depth. However, the syntype specimens illustrated by *Gooday et al. (2010*, Pl. 1*)* have polyhedral tests that are clearly different from the compressed tests of *C. crassa* and *C. patelliformis*. Additionally, the test of *P. bowmanni* includes a larger number of mica plates than either of these two species. Lastly, in *P. bowmanni,* sediment particles are agglutinated at the edges of mica flakes, and the size of agglutinated particles is much finer than *C. crassa* and *C. patelliformis*.

## CONCLUSIONS

We have established *C. crassa* sp. nov., a new monothalamous foraminiferal species from shallow water (<60 m) on the east coast of Korea. *C. crassa* displays typical morphological characteristics of the genus, namely a sandwich-like structure in which two (occasionally three) mica plates confine a ring-like formation composed of small mineral grains surrounding the cell body. Although it closely resembles the genotype *C. patelliformis* from the lower bathyal (~3,500 m depth) Portuguese margin, differences in size and the wide geographical and bathymetric separation between the type localities support our conclusion that these two species are distinct. Unfortunately, we failed to obtain SSU rDNA sequences from *C. crassa*. Further efforts will be made to remedy this lack of genetic data in order to clarify the phylogenetic relationship between *C. crassa* and *C. patelliformis,* as well as between both these species and members of the *Crithionina,* a genus that resembles *Capsammina* in a number of respects.

This new species extends the geographical distribution of the genus *Capsammina* as well as our knowledge of foraminiferal biodiversity around the Korean peninsula.

In particular, it increases the number of relatively robust monothalamid genera in the order Astrorhizida known from this region from seven to eight, and the number of astrorhiziid species from 11 to 12 (*Kim et al., 2016*). The monothalamids were largely excluded from *Kim et al. (2016)*'s otherwise comprehensive survey of Korean foraminiferal diversity. It seems likely that future investigations will reveal additional species belonging to this group of "primitive" foraminifera.

## ACKNOWLEDGEMENTS

We thank Raehyuk Jeong and Jisu Yeom for field work to collect samples on May 2016, Jaehyun Kim for field work to collect the new species on May 2018. We appreciate very much to academic editor, Joseph Gillespie, and two reviewers, Michael Kaminski and Anna Waśkowska for their critical comments on the earlier version of manuscript.

### Funding

This work was supported by the Ministry of Environment (MOE) of the Republic of Korea to the National Institute of Biological Resources (No. NIBR201839201), and by the Marine Biotechnology Program of the Korea Institute of Marine Science and Technology Promotion (KIMST, No. 20170431) funded by the Ministry of Oceans and Fisheries (MOF) to the Marine Biodiversity institute of Korea (MABIK). The funders had no role in study design, data collection and analysis, decision to publish, or preparation of the manuscript.

### Grant Disclosures

The following grant information was disclosed by the authors:
Ministry of Environment (MOE) of the Republic of Korea to the National Institute of Biological Resources: NIBR201839201.
Marine Biotechnology Program of the Korea Institute of Marine Science and Technology Promotion (KIMST): 20170431.
Ministry of Oceans and Fisheries (MOF).
Marine Biodiversity institute of Korea (MABIK).

### Competing Interests

The authors declare that they have no competing interests.

### Author Contributions

- Somin Lee conceived and designed the experiments, performed the experiments, analyzed the data, contributed reagents/materials/analysis tools, prepared figures and/or tables, authored or reviewed drafts of the paper, approved the final draft.
- Eric Armynot du Châtelet performed the experiments, analyzed the data, contributed reagents/materials/analysis tools, prepared figures and/or tables, authored or reviewed drafts of the paper, approved the final draft.

- Andrew J. Gooday analyzed the data, authored or reviewed drafts of the paper, approved the final draft.
- François Guillot performed the experiments, analyzed the data, contributed reagents/ materials/analysis tools, prepared figures and/or tables, approved the final draft.
- Philippe Recourt performed the experiments, analyzed the data, contributed reagents/ materials/analysis tools, prepared figures and/or tables, approved the final draft.
- Fabrizio Frontalini conceived and designed the experiments, analyzed the data, authored or reviewed drafts of the paper, approved the final draft.
- Wonchoel Lee conceived and designed the experiments, analyzed the data, prepared figures and/or tables, authored or reviewed drafts of the paper, approved the final draft.

## Data Availability

The raw data and measurements are available in Tables S1, S2 and S3. Tables S1 and S2 show mica plate measurements. Table S3 shows the result data of ESEM-EDS analyses of six specimens.

## New Species Registration

The following information was supplied regarding the registration of a newly described species:

Publication LSID: urn:lsid:zoobank.org:pub:3DDBFCFC-5927-41C4-9BB6-DDDD39DC22ED.

*Capsammina crassa* sp. nov. LSID:
urn:lsid:zoobank.org:act:712CADFE-DC11-4BA6-BDD4-A0EB1D80343D.

## Supplemental Information

Supplemental information for this article can be found online at http://dx.doi.org/10.7717/peerj.6642#supplemental-information.

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
