# Peer review of "The chemical composition of a new “mica sandwich” foraminiferal species from the East Coast of Korea: Capsammina crassa sp. nov"

_PeerJ, doi:10.7717/peerj.6642_

## Round 0.1 · original submission · Minor Revisions

Dear Dr. Lee and colleagues:

Thanks for submitting your manuscript to PeerJ. I have now received two independent reviews of your work, and as you will see, both are very favorable. Well done! Nonetheless, both reviewers raised some relatively minor concerns about the research, and areas where the manuscript can be improved. I agree with the reviewers, and thus feel that their concerns should be adequately addressed before moving forward.

In particular, please seriously consider deleting the dialogue about P. bowmanni being congeneric or provide a solid argument for this assertion.

Therefore, I am recommending that you revise your manuscript accordingly, taking into account all of the issues raised by the reviewers. I do believe that your manuscript will be ready for publication once these issues are addressed.

Good luck with your revision,

-joe

·

Basic reporting

The paper is clear, concise and a pleasure to read. It's a basic report of a newly-found species, with broader comparisons with the type species of the genus. The description has sufficient detail, and illustrations are excellent, showing the new species in different views.

Experimental design

Fine. No problems here.

Validity of the findings

I agree that this is likely to be a new species of Capsammina. The Sea of Japan is a semi-isolated body, and endemic species are expected. This is certainly the case with the Miocene, which is host to many endemic species.

The genus Capsammina was described as being sandwich-like, comprised of a doughnut-like ring of fine agglutinated grains sandwiched between two mica flakes. Kaminski (2014) placed it in the subfamily Psammosphaerinae, because it was described as possessing no obvious aperture. However, two images in Figure 2 bring this into question. In the side-view image (fig. 2-3) a hole (or at least a dimple) is visible. In top view (Fig. 2-1), the pavement of agglutinated grains (at 12 o'clock) seems to stem from the position of the hole. Agglutinated grains in the doughnut at this location appear to be mostly coarse, with less fine-grained material. Is this the aperture? Or is the connection through the more permeable (?thinner) wall at this location?

One thing I do not agree with is the inclusion (albeit speculative) of the species Psammosphaera bowmanni into the genus Capsammina. Capsammina was described as a "Mica Sandwich", i.e. two large plates cemented together by a ring of fine agglutinated material. I don't like the inclusion of P. bowmanni for two reasons: First, P. bowmani uses several flakes of mica to construct a polygonal "box". If we start separating out forms that build their test out of Mica, then we also need to describe a new genus for the species Reophax micaceus. Secondly, in P. bowmanni and in R. micaceus, the mica flakes are cemented at their edges by a matrix of finer agglutinated particles. This is not the case in Capsammina.

I know that in their original description of the genus, the authors mentioned that P. bowmanni may belong in Capsammina. No need to repeat this speculation in the current paper.

Additional comments

Good finding! I agree that this is a new species, and the illustrations are excellent.

However, The genus Capsammina was described as being sandwich-like, comprised of a doughnut-like ring of fine agglutinated grains sandwiched between two mica flakes. Kaminski (2014) placed it in the subfamily Psammosphaerinae, because it was described as possessing no obvious aperture. However, two images in Figure 2 bring this into question. In the side-view image (fig. 2-3) a hole (or at least a dimple) is visible. In top view (Fig. 2-1), the pavement of agglutinated grains (at 12 o'clock) seems to stem from the position of the hole. Agglutinated grains in the doughnut at this location appear to be mostly coarse, with less fine-grained material. Is this the aperture? Or is the connection through the more permeable (?thinner) wall at this location? Would you like to comment on this observation?

One thing I do not agree with is the inclusion (albeit speculative) of the species Psammosphaera bowmanni into the genus Capsammina. Capsammina was described as a "Mica Sandwich", i.e. two large plates cemented together by a ring of fine agglutinated material. By the way, Capsammina may also cement together shards of angular quartz - I have seen agglutinated doughnuts attached to the flat sides of angular quartz shards in my Arctic material. I don't like the inclusion of P. bowmanni for two reasons: First, P. bowmani uses several flakes of mica to construct a polygonal "box". If we start separating out forms that build their test out of Mica, then we also need to describe a new genus for the species Reophax micaceus. Secondly, in P. bowmanni and in R. micaceus, the mica flakes are cemented at their edges by a matrix of finer agglutinated particles. This is not the case in Capsammina.

I know that in their original description of the genus, the authors mentioned that P. bowmanni may belong in Capsammina. No need to repeat this speculation in the current paper.

·

Basic reporting

no comment

Experimental design

no comment

Validity of the findings

no comment

Additional comments

Very interesting nauscript, important because of better knowledge of foraminifers and their taxonomic diversity. The species that is described is spectacular because the test structure.
The article is very well written, a new species is widely documented and well-illustrated.
I have only minor comments that could be taken into consideration.
It is worth considering if the abstract requires information about other species of Capsammina (the last two sentences).
In conclusions (the last paragraph) give data about biodiversity in the area of Korea peninsula. In principle, this topic is poorly discussed in the content of the manuscript.

---

## Round 0.2 · accepted · Accept

Dear Dr. Lee and colleagues:

Thanks for revising your manuscript based on the minor concerns raised by the reviewers. I now believe that your manuscript is suitable for publication. Congratulations! I look forward to seeing this work in print, and I anticipate it being an important resource for the groups studying monothalamous foraminifera and overall deep ocean microbial diversity. Thanks again for choosing PeerJ to publish such important work.

Best,

-joe

#